# FILTER BEFORE PLUG: ONE-FOR-ALL FRAMEWORK FOR COVARIATE-AWARE FORECASTING WITH TIME SERIES FOUNDATION MODELS

## ABSTRACT

Time series forecasting plays a critical role in numerous real-world applications. Recent advances in Time Series Foundation Models (TSFMs) have achieved strong performance by modeling historical dependencies; however, they frequently neglect the impact of exogenous covariates. Existing methods either train from scratch, losing the advantages of TSFMs, or design plugin modules that are tightly coupled with specific architectures. To address these limitations, we propose FLUG, a One-for-All framework where independently trained modules complement TSFMs. We design an Endogenous Series Filter (EFit) module guided by the Hurst Exponent to separate exogenous components from the time series, thereby enabling TSFMs to focus on modeling and forecasting endogenous patterns. In parallel, we introduce a Covariate Plugin (CPin) module that employs Multi-Scale Patchify fusion and a Causal-Aware Masking strategy based on Gradient Reversal Layer to capture the exogenous information of the target variable. By decomposing endogenous and exogenous dependencies, FLUG enables integration of covariate information across a variety of TSFMs. To supplement existing publicly available covariate time series data, we curate and release four additional datasets. Extensive experiments on real-world business and supplementary data demonstrate the framework's effectiveness and scalability.

## 1 INTRODUCTION

Time series forecasting(TSF) tasks aim to predict the future based on given historical series. Accurate forecasting supports decision-making processes, making TSF crucial in various domains such as energy Qiu et al. (2024), finance Mei et al. (2025), and environment Tian et al. (2025).

In recent years, Foundation Models have achieved remarkable success in fields like natural language processing Touvron et al. (2023); Brown et al. (2020) and computer vision Dosovitskiy et al. (2021); Liu et al. (2021). By utilizing various data and large-scale parameter architectures, these models exhibit outstanding generalization abilities and demonstrate impressive performance even in zero-shot tasks. Inspired by these achievements, researchers have begun to explore the potential of time series foundation models (TSFMs) Das et al. (2024); Ansari et al. (2024); Wang et al. (2025). Existing TSFMs focus on modeling historical series to capture trends, periodicity, and specific patterns within the data for forecasting future values. This approach operates under the fundamental assumption that time series exhibit complete dependence on historical data, with recurring or similar characteristics consistently appearing over time.

However, the time series data in real-world production often relies not only on historical dependencies but also on external interventions by exogenous variables or covariates. For example, in photovoltaic power generation, while daily power generation can be inferred from historical power patterns, it directly depends on the real-time irradiation level. Currently, there are two paradigms for incorporating exogenous variables into time series forecasting models. Most methods Wang et al. (2024b); Liu et al. (2025a), as shown in Figure 1(a), adopt an end-to-end training paradigm from scratch. This approach sacrifices the powerful forecasting capabilities of TSFMs. Methods like ChronosX Pineda-Arango et al. (2025) introduce plugin modules into TSFMs to enable exogenous variables modeling, as shown in Figure 1(b). Existing covariate plugins are designed around

forecasting capabilities of TSFMs, merely supplementing them with covariate information, making these plugins inherently dependent on specific TSFMs.

This motivates us to design independently trainable modules that can easily adapt to various TSFMs without fine-tuning, as shown in Figure 1(c). To independently train this module, it is necessary to decompose the dependency patterns in time series data, separating endogenous and exogenous dependency information. The TSFM is utilized to forecast endogenous dependency information, while the plugin serves to supplement exogenous covariate information, as illustrated in Figure 1(c). However, it is a challenge to extract endogenous information from the target variable time series. Existing time series decomposition methods, such as DFT Zhou et al. (2022) and wavelet decomposition Chen et al. (2025), do not

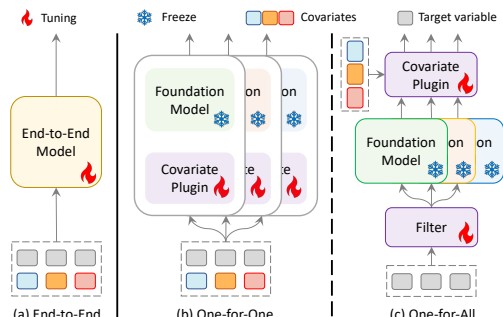

Figure 1: Illustration of existing methods for TSF with covariates.

explicitly decompose the endogenous information in time series data. Meanwhile, there are time delays and multi-scale impacts between different variables due to the dynamic nature of time series data. This means that not all covariates exhibit causal relationships with the target variable at every time step. Such challenges make it difficult to accurately capture and model the causal relationships between covariates and the target variable.

To address these challenges, we propose FLUG, a One-for-All framework realized by independently training two modules alongside the TSFMs. We first propose an **Endogenous Series Filter** module guided by Hurst Exponent Mandelbrot & Wallis (1969) constraints, which extracts endogenous information with historical dependencies from the target variables. The Hurst Exponent quantifies the long-term memory of a time series, where a higher value indicates stronger historical dependency. Fractional Brownian Motion (FBM) Mandelbrot & van Ness (1968), governed by the Hurst Exponent, exhibits stationarity and multi-scale self-similarity. Through a multi-scale discriminator, the representation of target variables is reconstructed into a time series that shares similar multi-scale natures with the generated FBM. Simultaneously, a Hurst Exponent-based loss is applied to enforce the reconstructed time series to exhibit the desired historical dependency. Through this reconstructed time series within this module, we filter out the historical dependency components from the target time series. In addition, we propose a causal-based multi-scale **Covariate Plugin** module that captures exogenous information dependent on covariates within the target variable. In causal inference, covariates that significantly influence the target variable are often termed as its causes Pearl (2022). We design a causal-aware masking module based on the Gradient Reversal Layer (GRL), which identifies the most impactful covariates to determine the causes of the target variable. Additionally, we employ time-aware attention and a multi-scale patch fusion strategy to model temporal lag effects and multi-scale correspondences between covariates and the target variables. By training the Filter and Plugin modules independently from the TSFMs, we achieve a One-for-All framework that integrates covariate information.

Specifically, our contributions can be summarized as follows:

- We propose FLUG, a novel paradigm of the One-for-All framework with Time Series Foundation Models, achieving a training paradigm independent of the TSFMs.

- We design a novel Endogenous Series Filter module based on the Hurst Exponent, capable of extracting historically dependent components from time series. The extracted series can then be modeled by TSFMs to capture temporal dependencies for TSF tasks.

- We develop a Covariate Plugin Module that utilizes causal-aware masking to capture causal relationships between covariates and the target variable. Furthermore, leveraging time-aware attention and multi-patch fusion, we model the time delay and multi-scale relationships between covariates and the primary variable.

- We release a real-world dataset based on business scenarios and conduct extensive experiments on both proprietary business data and publicly available covariate datasets to validate the framework's effectiveness and scalability.

## 2 RELATED WORK

### 2.1 TIME SERIES FOUNDATION MODELS

Foundation models pre-trained on large-scale datasets have achieved remarkable success in natural language processing Lewis et al. (2020); Raffel et al. (2020); Touvron et al. (2023) and computer vision Dosovitskiy et al. (2021); Liu et al. (2021). In recent years, time-series foundation models Woo et al. (2024); Liu et al. (2025b); Wang et al. (2024a) have also shown outstanding progress. These pre-trained models have demonstrated strong performance across various downstream forecasting tasks, excelling in few-shot fine-tuning and even zero-shot scenarios. However, due to the heterogeneity of time-series data across different domains, existing foundation models often adopt a channel-independent training strategy Nie et al. (2023) to facilitate unified training. While this approach focuses on modeling dependencies within individual time series, it neglects the interactions and relationships between different variables. This limitation can lead to poor performance on tasks where cross-variable interactions play a critical role.

### 2.2 TIME SERIES FORECASTING WITH COVARIATES

Modeling the relationship between covariates and target variables differs fundamentally from modeling inter-channel dependencies. Covariates typically refer to future-known variables that, based on prior knowledge, are related to the target variables. Existing methods for covariate modeling can also be broadly categorized into two types. The first category integrates covariate information directly into forecasting models through end-to-end training. For example, TimeXer Wang et al. (2024b) employs cross-attention between covariate and target variables to effectively incorporate covariate information. Similarly, Timer-XL Liu et al. (2025a) uses causality-based time attention to model the relationships between covariates and target variables. While these methods embed covariate information during training, they often suffer from limited generalization because they do not fully exploit the predictive power of pre-trained TSFMs. The second category focuses on extending TSFMs by designing covariate integration plugins, allowing TSFMs initially trained on univariate time-series data to leverage covariates effectively. For example, ChronosX Pineda-Arango et al. (2025) introduces plugins to enable covariate fusion, which augments the capabilities of TSFMs to handle covariates. However, such plugins are usually tightly coupled with specific TSFMs, making them less flexible and harder to adapt to different TSFMs.

To address these limitations, we propose FLUG, a novel framework that can be trained independently of any specific TSFM. This independence allows FLUG to seamlessly adapt across various TSFM architectures with minimal effort, enabling rapid integration of covariate information.

## 3 METHODOLOGY

In times series forecasting with covariates, the objective is to predict $\mathbf{Y}_{T+1:T+H} \in \mathbb{R}^{H \times 1}$ by utilizing both context series $\mathbf{Y}_{1:T} \in \mathbb{R}^{T \times 1}$ and covariates $\mathbf{X}_{1:T+H} \in \mathbb{R}^{(T+H) \times N}$ where the $N$ is the numbers of covariates. Time Series Foundation Models (TSFMs) are designed to capture the temporal dependencies within time series for forecasting, but are weak at modeling the correlation between the target variable and covariates. The Covariate-aware adaptation method incorporates a covariate modeling module into TSFMs, enabling TSFMs to leverage covariate information. The formulation of this paradigm can be expressed as follows: Given a TSFM $f(\cdot)$, the objective is to build a new forecasting model based on the pre-trained model:

$$\widetilde{\mathbf{Y}}_{T+1:T+H} = g \circ f(\mathbf{Y}_{1:T}, \mathbf{X}_{1:T+H}), \tag{1}$$

where $g(\cdot)$ is the adaptation plugin, and $g \circ f(\cdot)$ is the composition model after adaptation.

### 3.1 OVERALL ARCHITECTURE

We propose FLUG, as shown in Figure 2, a novel paradigm of the One-for-All framework with Time Series Foundation Models (TSFMs), which enhances various TSFMs with the ability to model covariates for auxiliary forecasting through a set of independently trained, plug-and-play modules.

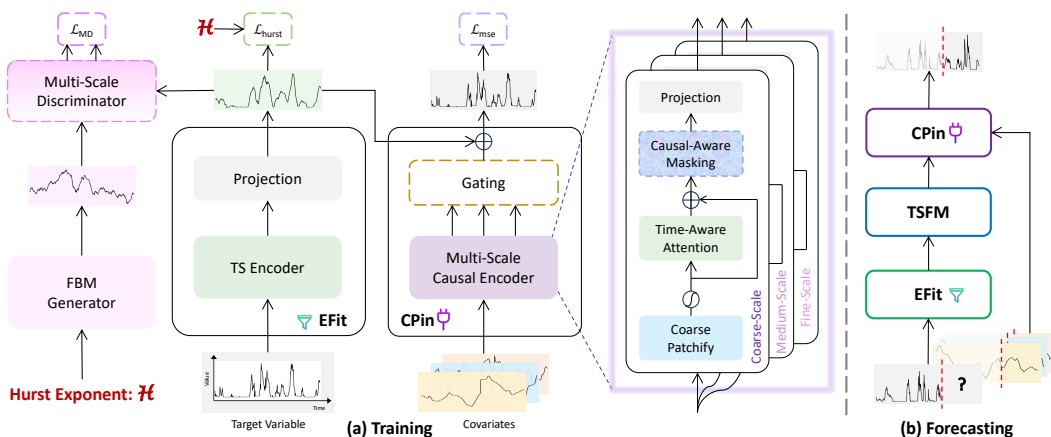

Figure 2: Overall of FLUG.

At the training stage, we input the target variable and covariates into the Endogenous Series Filter (EFit) and Covariate Plugin (CPin) modules, respectively. We use the EFit to extract the endogenous dependency information from the target variable. We generate FBM with a given Hurst Exponent and use a Time Series Encoder to extract target representations. A Multi-Scale Discriminator ensures the extracted series captures endogenous multi-scale dependencies, while a Hurst Loss constrains its Hurst Exponent. Meanwhile, the CPin module captures multi-scale relationships between covariates and the target through patchification and gating mechanisms.. From the perspective of causal root reasoning, we design a data-adaptive masking strategy based on a gradient reversal layer (GRL) to capture the causal relationship between the covariates and the target variable. Finally, we fuse the endogenous information extracted from the target variable with the exogenous information from the covariates to obtain the complete series.

At the forecasting stage, we use the trained EFit to filter the context series of the target variable and use the TSFMs to perform zero-shot forecasting on the filtered series. Through the trained CPin module, the exogenous information from future covariate series is supplemented into the TSFM's prediction results, resulting in the final forecast series.

## 3.2 ENDOGENOUS SERIES FILTER

Time series often exhibit periodic and regular patterns—stable endogenous information at a macroscopic scale, which makes time series forecasting possible. The outstanding performance of TSFMs on a wide range of forecasting tasks has already demonstrated their effectiveness in capturing steady-state endogenous information within time series data. However, in certain complex scenarios, time series data can be heavily influenced by various external factors. To extract time-dependent steady-state endogenous information and better utilize the modeling capabilities of TSFMs for such information, we propose an EFit module along with an endogenous information extraction strategy.

Firstly, we use a TS encoder to extract representations from the target time series data and generate a series. A Hurst-based loss constrains the extracted series to be temporally dependent and endogenous. Meanwhile, we generate FBM based on a given Hurst exponent. Subsequently, a Multi-Scale Discriminator extracts endogenous information from the TS representations to approximate the multi-scale characteristics of FBM. This yields a EFit module capable of extracting endogenous information from the target time series.

**Hurst-based Loss.** The Hurst Exponent Mandelbrot & Wallis (1969) is used to measure the temporal dependence of time series. When the Hurst Exponent exceeds 0.5, the series shows strong temporal dependence; a detailed description of the Hurst Exponent is provided in the Appendix C.1. Such sequences tend to exhibit more pronounced trends and periodicity, resulting in higher predictability. In the generated time series $\hat{\mathbf{Y}}$, there is a power-law relationship between the time scale $t$ and the variance of the series fluctuations $\sigma^2 = \text{Var}[\hat{\mathbf{Y}}]$ that is related to the Hurst Exponent $H$.

The power-law relationship can be formalized as:

$$\sigma^2(t) \propto t^{2H}. \tag{2}$$

By taking the logarithm of the above formula, we can derive:

$$\ln \sigma^2(t) = 2H \ln t + C, \tag{3}$$

where $C$ is a constant. Temporal dependence requires a power-law relationship to hold across multiple time scales. Hence, based on the least-squares method, we construct the Hurst-based loss defined as:

$$\mathcal{L}_{\mathbf{hurst}} = \left| \frac{\sum_{k=1}^{K} \left( \ln t_k - \ln t \right) \left( \ln \sigma_k^2 - \ln \sigma^2 \right)}{\sum_{k=1}^{K} \left( \ln t_k - \ln t \right)^2} - 2H \right|, \tag{4}$$

where $t_k$ represents the $k$-th time scales, and $\ln t = \frac{1}{K} \sum_{k=1}^{K} \ln t_k$ represents the average across the $k$ scales. Likewise $\sigma_k^2$ represents the variance of different series fluctuations, and $\ln \sigma^2 = \frac{1}{K} \sum_{k=1}^{K} \ln \sigma_k^2$.

**Multi-scale Discriminative Loss.** Using Hurst-based loss can preliminarily ensure the temporal dependence of the extracted sequence, but constraints based on a single metric are not sufficient to guarantee the temporal dependence of the extracted sequence. Fractional Brownian Motion (FBM) is a sequence based on the Hurst Exponent that satisfies temporal dependence across multiple time scales. With an FBM Generator, an FBM $M$ can be obtained from a Hurst Exponent greater than 0.5. To further constrain the temporal dependence of the generated sequence, we expect the generated sequence to be multi-scale, similar to FBM. Specifically, we obtain the multi-scale decomposition information of FBM and the generated sequence $\mathbf{Y}$ through wavelets. A macro-multiscale approximation is performed by a Moment-based loss as:

$$\mathcal{L}_{\mathrm{moment}} = \left\| \Psi(\hat{\mathbf{Y}}) - \Psi(\mathbf{M}) \right\|_2^2. \tag{5}$$

where $\Psi(\cdot)$ denotes multi-scale statistical moment features. To further constrain the multi-scale approximation of FBM and the generated sequence at the microscopic scale, we designed a Multi-Scale Discriminator $D(\cdot)$. The discriminator is used to extract multi-scale features of both the FBM and the generated sequence, and they are constrained by a Feature-based loss, defined as:

$$\mathcal{L}_{\mathrm{feat}} = \sum_{k=1}^{K} \left\| D(\hat{\mathbf{Y}}) - D(\mathbf{M}) \right\|_2^2. \tag{6}$$

By applying the constraints of the two losses above, we are able to train the EFit module that extracts the endogenous information sequence within the target sequence. The full loss is as follows:

$$\mathcal{L}_{\mathrm{MD}} = \alpha \mathcal{L}_{\mathrm{moment}} + (1 - \alpha) \mathcal{L}_{\mathrm{feat}}, \tag{7}$$

where $\alpha$ is the given hyperparameter.

### 3.3 COVARIATES PLUGIN

The dynamic nature of time series data leads to time delays and multi-scale causal relationships between covariates and the target variable, making it challenging to model these relationships. In scenarios with multiple covariates, the impact of different covariates on the main variable can vary across different time periods. To enable data-adaptive multi-scale modeling of the dynamic causal relationships between various covariates and the target variable, we design the Covariates Plugin (CPin) module.

Firstly, we employ a Multi-Scale Causal Encoder to model the covariate series. Covariates are segmented into patches of three different scales, and a Causal-Aware Masking strategy is utilized to retain the covariates that exert the greatest influence on the target variable, thereby fully exploiting the causal relationships between the target variable and the covariates. Then, we use the Gating mechanism to fuse the series obtained at different scales, resulting in an exogenous information series for the target variable. Finally, the exogenous information series is added to the endogenous information series extracted by the EFit module to obtain the final output.

**Multi-Scale Modeling.** The influence of covariates on the main variables is multi-scale. Taking a photovoltaic power generation scenario as an example, irradiance impacts power generation instantaneously, while humidity effects often manifest over longer time scales. When multiple covariates exist simultaneously, multi-scale modeling and fusion are essential. Therefore, we implement multi-scale modeling through a Multi-Scale Patchify strategy and a Gating mechanism. Previous work Nie et al. (2023) has highlighted the significance of patch strategies in sequential modeling. We capture information at three scales—coarse, medium, and fine—by using patches of three sizes to map covariates $\mathbf{X}^N$, where $N$ is the number of covariates, to the corresponding target variable at different scales. Position embeddings are added to the three patch embeddings, and time-aware attention models the temporal relationships across the same scale patches. By mapping the $N$ covariates $\mathbf{X}^N$ to the corresponding times of the target variable $\mathbf{Y}$, we obtain the exogenous information sequences for the target variable at three scales. After fusing the three sequences with a gating mechanism, they are added to the endogenous information sequence filtered by the EFit module, yielding the final estimated target variable sequence $\widetilde{\mathbf{Y}}$. The mean squared error (MSE) loss $\mathcal{L}_{\mathbf{mse}}$ between the estimated sequence $\widetilde{\mathbf{Y}}$ and the original target variable $\mathbf{Y}$ is constructed as:

$$\mathcal{L}_{\mathbf{mse}} = \left\| \widetilde{\mathbf{Y}} - \mathbf{Y} \right\|_2^2. \tag{8}$$

Through the MSE loss, we have constructed the multiple-scale relationship between covariates and the target variable.

**Causal-Aware Masking.** The impact of covariates on the target variable extends beyond multi-scale challenges, and establishing their causal relationship with the target variable is equally crucial. If perturbing one variable leads to a significant change in another, the perturbed variable is likely the cause of the latter. Based on this, we designed a Covariate-Aware Masking strategy, as shown in the Figure 3. By initially masking the representations of these covariates, we can obtain a more robust relation extractor. The Gradient Reversal Layer (GRL) Ganin & Lempitsky (2015) operates as an identity function during forward propagation, while reversing the gradients during backpropagation. This makes the mask generator's objective opposite to the feature ex-

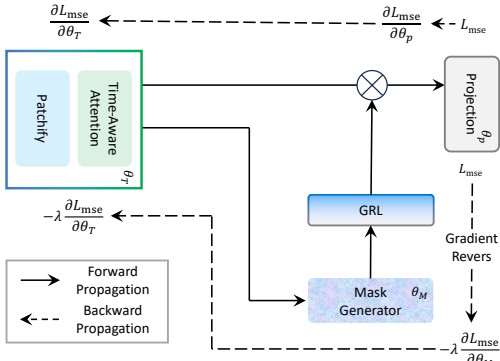

Figure 3: Illustration of Causal-Aware Masking.

tractor, as it seeks to mask certain covariates. To satisfy the overall optimization objective while having to mask certain covariates, the mask generator identifies and preserves the most influential covariates during training. This way, we obtain a dynamic masking strategy that can mask non-causal covariates while preserving causal covariates.

## 4 EXPERIMENTS

### 4.1 EXPERIMENTAL DATASETS

**Open-source Datasets.** To verify the generality of our method, we selected some open-source datasets containing covariates, including PDB Yeafi (2021), GFC14 Hong et al. (2016), BDG2Cockatoo Lago et al. (2020a), and EPFPJM Lago et al. (2020b). These datasets generally come from public data and algorithm competitions, making the authenticity of the data difficult to verify. These four selected datasets are sampled at a frequency of one observation per hour.

**Ningxia, Jinlin Photovoltaic and Wind Power Datasets.** Most publicly available datasets contain few covariates and often suffer from mismatched or aggregated information, limiting their usefulness for exploring covariate effects. To provide supplementary data and improve quality, we processed and released an industrial dataset. We process and release datasets on power generation efficiency from several photovoltaic and wind power stations located in Ningxia province and Jilin

Table 1: Details of Datasets

| Region | Type | Freq. | Subsets | Points (million) Obs. | Points (million) Covar. | Covar. |
|---|---|---|---|---|---|---|
| Jilin | PV | | 22 | 14.6 | 351.2 | 24 |
| Jilin | Wind | | 20 | 13.9 | 334.5 | 24 |
| | | 15-min | | | | |
| Ningxia | PV | | 58 | 14.8 | 357.6 | 24 |
| Ningxia | Wind | | 40 | 10.1 | 244.3 | 24 |

province, China. We select the weather data forecast for the most recent report time from meteorological stations as the weather data for that moment. According to the station location information, we use the corresponding local weather forecast data as covariates. Detailed information is shown in the following Table 1. The sampling frequency of the generation efficiency and weather data is denoted under Freq. To avoid the impact of prolonged missing segments on the analysis, we split the data at both ends of each long missing segment. A total of 140 sub-datasets were extracted from the four data types, with each sub-dataset containing 24 covariate variables and one target variable. Altogether, the four datasets comprise approximately 1.3 billion data points.

## 4.2 EXPERIMENTAL SETTINGS

We conduct a one-day-ahead prediction using one week of historical data. For hourly public datasets, this corresponds to 24 future points from 168 past observations, while for the 15-minute Ningxia and Jilin datasets, we predict 96 future points from 672 past observations. We selected some of the latest and classic state-of-the-art algorithms as baselines, including TSFMs such as TimesFM Das et al. (2024) and Chronos Ansari et al. (2024), end-to-end models considering covariate modeling like Timer-XL Liu et al. (2025a) and TimeXer Wang et al. (2024b), and multivariate relationship-aware models such as iTransformer Liu et al. (2024a) and channel-independent PatchTST Nie et al. (2023). All the experiments are implemented with PyTorch on four PPU-ZW810E 90GB GPUs. Experimental results are averaged over five runs with different seeds.

## 4.3 EXPERIMENTAL RESULTS

Table 2: Model performance on four datasets (PDB, GFC14, BDG2Cockatoo, EPFPJM)

| Model | Metric | PDB | GFC14 | BDG2Cockatoo | EPFPJM | Avg. |
|---|---|---|---|---|---|---|
| **FLUG**+TimesFM | MSE | 0.067 | **0.119** | **0.134** | **0.077** | **0.099** (-3.9%) |
| **FLUG**+TimesFM | MAE | 0.193 | **0.233** | **0.264** | **0.176** | **0.216** (-3.2%) |
| **FLUG**+Chronos | MSE | **0.061** | **0.109** | **0.102** | **0.074** | **0.087** (-5.1%) |
| **FLUG**+Chronos | MAE | **0.173** | **0.226** | **0.234** | **0.173** | **0.202** (-4.6%) |
| TimesFM | MSE | 0.093 | 0.198 | 0.176 | 0.085 | 0.138 |
| TimesFM | MAE | 0.206 | 0.302 | 0.300 | 0.183 | 0.248 |
| Chronos | MSE | 0.078 | 0.192 | 0.164 | 0.086 | 0.130 |
| Chronos | MAE | **0.173** | 0.300 | 0.296 | 0.183 | 0.238 |
| Timer-XL | MSE | 0.146 | 0.276 | 0.247 | 0.089 | 0.190 |
| Timer-XL | MAE | 0.270 | 0.378 | 0.367 | 0.187 | 0.301 |
| TimerXer | MSE | 0.433 | 0.567 | 1.054 | 0.088 | 0.536 |
| TimerXer | MAE | 0.530 | 0.499 | 0.755 | 0.188 | 0.493 |
| iTransformer | MSE | 0.082 | 0.159 | 0.160 | 0.097 | 0.125 |
| iTransformer | MAE | 0.200 | 0.280 | 0.297 | 0.197 | 0.244 |
| PatchTST | MSE | **0.065** | 0.159 | 0.196 | 0.106 | 0.132 |
| PatchTST | MAE | **0.173** | 0.276 | 0.320 | 0.209 | 0.245 |

For TSFMs models, we use pre-trained models to perform zero-shot prediction on the test set. For end-to-end models, we conduct separate training and prediction on different datasets. Our method is trained independently on the training set and adapted to two TSFMs models, respectively, for prediction, with the TSFMs model parameters frozen.

We use MSE and MAE as evaluation metrics, where lower values indicate better performance. For the four public energy datasets, each containing multiple sub-datasets, we compute the met-

Table 3: Model performance on energy datasets (Jilin and Ningxia).

| Model | Metric | Jilin | | Ningxia | | Avg. |
|---|---|---|---|---|---|---|
| | | PV | Wind | PV | Wind | |
| **FLUG**+TimesFM | MSE | 0.279 | **0.762** | **0.213** | **1.452** | **0.676**(-11.7%) |
| | MAE | 0.302 | **0.653** | **0.264** | **0.662** | **0.470**(-2.5%) |
| **FLUG**+Chronos | MSE | **0.247** | 0.786 | **0.213** | **1.477** | **0.681**(-22.3%) |
| | MAE | **0.275** | **0.679** | 0.268 | **0.703** | **0.481**(-5.4%) |
| TimesFM | MSE | 0.420 | 0.992 | 0.281 | 1.478 | 0.793 |
| | MAE | 0.311 | 0.713 | 0.271 | 0.704 | 0.495 |
| Chronos | MSE | 0.302 | 1.133 | 0.247 | 1.935 | 0.904 |
| | MAE | **0.235** | 0.782 | 0.271 | 0.850 | 0.535 |
| Timer-XL | MSE | 0.291 | 1.032 | 0.241 | 1.761 | 0.831 |
| | MAE | 0.302 | 0.795 | 0.288 | 0.853 | 0.560 |
| TimerXer | MSE | 0.695 | 1.155 | 0.477 | 1.876 | 1.051 |
| | MAE | 0.621 | 0.839 | 0.519 | 0.898 | 0.719 |
| iTransformer | MSE | 0.308 | 0.966 | 0.275 | 1.931 | 0.870 |
| | MAE | 0.282 | 0.755 | 0.322 | 0.873 | 0.558 |
| PatchTST | MSE | **0.272** | 0.899 | 0.254 | 1.792 | 0.804 |
| | MAE | 0.298 | 0.726 | 0.309 | 0.832 | 0.541 |

rics per sub-dataset and average them. Results are shown in Tables 2 (public datasets) and 3 (Ningxia and Jilin). Blue bold indicates the best, underlined bold the second-best, and red bold shows improvements from applying FLUG. Overall, TSFMs exhibit stronger generalization and prediction ability than end-to-end models, with zero-shot TSFMs often outperforming fully trained models, demonstrating their necessity. FLUG consistently improves TSFMs. On the four public datasets, Chronos+FLUG achieves the best performance, with TimesFM+FLUG ranking second on most datasets. On the Ningxia and Jilin energy datasets, TimesFM+FLUG often ranks first, while Chronos+FLUG ranks second, demonstrating FLUG's universal value and practical feasibility. These results also highlight that different TSFMs suit different domains and underscore the importance of comprehensive covariate modeling for accurate time series forecasting.

Table 4: Performance of Other Contributions

| | baseline | wo/ Hurst-based Loss | wo/ MD Loss | wo/ Masking |
|---|---|---|---|---|
| MSE | **0.762** | 0.787(+2.5%) | 0.862(+10.0%) | 0.819(+5.7%) |
| MAE | **0.653** | 0.664(+1.1%) | 0.681(+1.7%) | 0.663(+1.0%) |

## 4.4 ABLATION STUDY

We conduct ablation studies on the EFit and CPin modules to verify their effectiveness. We use TimesFM as the TSFM part within the FLUG framework for our experiments. All experiments were conducted on 20 datasets of Jilin wind power, and we calculated the average MSE and MAE across each dataset as the final result.

**Analysis of Multi-Scale Patchify.** To demonstrate the necessity of the multi-scale modeling strategy, we experiment with multiple scales to investigate their impact on prediction results. We test modeling with a single patch length as well as with up to five different patch lengths on this dataset, and record the MSE and MAE for each method. The results are shown in the Figure 4. The numbers on the x-axis indicate the number of patch lengths used for modeling, ranging from a single patch length to multiple scales: [32], [16, 32], [4, 16, 32], [4, 8, 16, 32], and [4, 8, 16, 32, 48]. Overall, multi-scale modeling significantly improves model performance. As the number

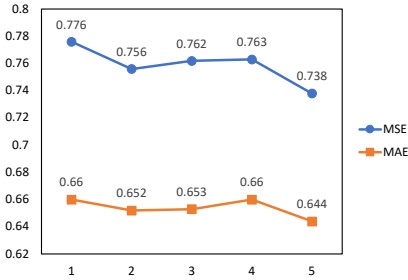

Figure 4: Performance of Multi-Scale Patchify.

of modeling scales increases, both MSE and MAE show a downward trend, and the model achieves the best results when five scales are used. This not only demonstrates the effectiveness of our proposed multi-scale modeling and gated fusion strategy, but also highlights the importance of multi-scale modeling for time-series data.

**Analysis of Other Contributions.** We design experiments to compare other innovative points in the paper. We separately verify the necessity of the Hurst-based Loss and Multi-scale Discriminative Loss used for training the EFit module, as well as the importance of the Causal-Aware Masking mechanism in the CPin module. The specific experimental results are shown in the Table 4. The baseline shows results of the complete model, while green values indicate performance drops after removing corresponding components. It can be seen from the table that the best results are achieved when the complete method is used. After removing the MD Loss, the model performance showed severe degradation on both MSE and MAE metrics. This experiment verifies the important role of our proposed components in endogenous information extraction from target variables and covariate information fusion.

### 4.5 ADDITIONAL EXPERIENCE

**Analysis on endogenous series.** We conduct experiments using data from Jinlin wind power, as shown in Figure 5. Figure 5(a) shows the original series with sharp fluctuations caused by multiple covariates, making prediction difficult. Figure 5(b) presents the endogenous series from our EFit module, which is smoother with subtler variations, similar to the generated FBM in Figure 5(c). Calculating the Hurst Exponent of the original series, endogenous series, and FBM shows that the EFit yields an endogenous series with the Hurst Exponent $> 0.5$. The endogenous series retains original patterns but is more stable, easing TSFM modeling and prediction.

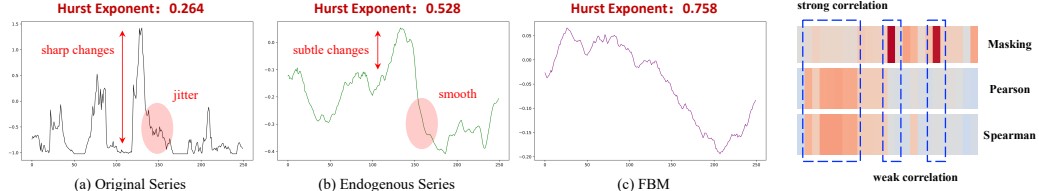

Figure 5: Visualization of the endogenous series.

Figure 6: Visualization of Masking.

**Analysis on Causal-Aware Masking.** We analyze the proposed Causal-Aware Masking using the Jilin wind power dataset, and the results are shown in Figure 6. We mask variable representations, with higher mask ratios shown as darker heatmap colors. We compute Pearson and Spearman coefficients between covariates and the target as baselines, with darker colors indicating stronger correlations. As shown in the Figure 6, variables with stronger correlations have lower mask ratios, while variables with weaker correlations are masked at higher proportions. This confirms that our method effectively identifies and preserves important covariates for the target variable, capturing the causal relationships between covariates and the target variable.

## 5 CONCLUSION

This paper presented FLUG, a One-for-All framework that enhances Time Series Foundation Models (TSFMs) with exogenous covariates. By disentangling endogenous and exogenous dependencies, FLUG employs a Hurst-guided Endogenous Series Filter module to extract endogenous patterns and a Covariate Plugin module with Multi-Scale Patchify fusion and Causal-Aware Masking to capture exogenous information of the target variable. To mitigate the limitations of existing covariate time series data, we further curated and released four supplementary datasets. Extensive experiments on real-world business and supplementary data demonstrate the effectiveness and scalability of our framework.

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

## A DETAILS OF DATASETS

### A.1 DETAILS OF OPEN-SOURCE DATASETS

Our study utilizes 4 open-source datasets. Table 5 provides detailed information on each dataset, including target variables, covariates, and sources, while Table 4 summarizes key dataset statistics. Specifically, several electricity load forecasting datasets—EPF-PJM, BDG-2 Cockatoo, GFC14, and PDB are included. EPF contains day-ahead electricity prices from five major power markets with load forecasts and wind generation as covariates Lago et al. (2020b). BDG-2 Cockatoo, collected from Cornell University, includes air temperature as a covariate Lago et al. (2020a). GEF14, spanning seven years of load series from the Global Energy Forecasting Competition 2014, provides the averaged temperature from the raw temperature series Hong et al. (2016). PDB comprises historical electric power load data with air temperature as a covariate Yeafi (2021).

Table 5: Dataset Descriptions

| Dataset Name | Descriptions | Covariates | Source |
|---|---|---|---|
| EPF | Day-ahead electricity prices from five major power markets: PJM | load forecasts, wind generation | Lago et al. (2020b) |
| BDG-2 Cockatoo | BDG-2 dataset collected from Cornell University. | air temperature | Lago et al. (2020a) |
| GFC14 | Seven years of load series data from the Global Energy Forecasting Competition 2014 | Averaged temperature from the raw 25 temperature data series | Hong et al. (2016) |
| PDB | The years of PDB electric power load history data from the Kaggle data competition. | air temperature | Yeafi (2021) |

### A.2 DETAILS OF NINGXIA, JINLIN PHOTOVOLTAIC AND WIND POWER DATASETS

We processed and released an industrial dataset on power generation efficiency from photovoltaic and wind power stations in Ningxia and Jilin, China. We use the latest local weather forecasts as covariates based on station locations. Table 1 provides detailed information, with sampling frequency indicated under Freq. Long missing segments are split at both ends to reduce their impact. In total, 140 sub-datasets were extracted, each containing 24 covariates and one target, comprising roughly 1.3 billion data points. The meanings of the weather covariates are listed in the Table 6.

## B COMPARED METHODS

Here, we present an overview of the baseline methods used in our experiments, emphasizing their methodological frameworks and approaches to handling covariates. We examine each method with respect to how it incorporates homogeneous covariates, highlighting the strengths and limitations in capturing external dependencies.

### B.1 PRETRAINED METHOD

**TimesFM Das et al. (2024).** TimesFM adopts a decoder-only Transformer architecture. In tokenization, the series undergoes preprocessing, fixed-length patching, and normalization using mean and standard deviation. Each patch is enriched with mask features and mapped into the embedding space, optionally with positional encodings. Contextual representations are learned through

Table 6: Details of Covariates

| Variable | Description | Unit |
|---|---|---|
| T | Temperature | °C |
| momf | Momentum flux | |
| sin_direction32 | Wind direction at 30m | degrees |
| ws30 | Wind speed at 170m | m/s |
| ws31 | Wind speed at 100m | m/s |
| ws32 | Wind speed at 30m | m/s |
| ws10 | Wind speed at 10m | m/s |
| ws10s | Wind speed at 10m | m/s |
| sin_direction30 | Wind direction at 170m | degrees |
| sin_direction31 | Wind direction at 100m | degrees |
| sin_dir10 | Wind direction at 10m | degrees |
| sin_dir10s | Wind direction at 10m | degrees |
| mslp | Mean sea level pressure | hPa |
| clc | Fraction of clouds [0-1] | |
| senf | Sensible heat flux | $W/m^2$ |
| latf | Latent heat flux | $W/m^2$ |
| swr | Shortwave radiation | $W/m^2$ |
| lwr | Longwave radiation | $W/m^2$ |
| ps | Surface pressure | hPa |
| prt | Total precipitation (PRT) | mm |
| prl | Large-scale precipitation | mm |
| prc | Convective-scale precipitation | mm |
| T2m | 2m temperature | °C |
| RH2m | Humidity | % |

multi-layer self-attention, while the decoder generates future values in an auto-regressive manner. The model outputs both mean and quantile forecasts, which are de-normalized to the original scale. Additionally, TimesFM incorporates frequency-based conditioning and hybrid-frequency modeling to enhance multi-scale forecasting.

**Chronos-Bolt Ansari et al. (2024).** Chronos-Bolt is built on a T5-based encoder–decoder architecture. The input time series is first processed with instance normalization and then divided into patches, accompanied by its mask; these two streams are concatenated before embedding. An optional [REG] token can be included to enable regression-style outputs. The encoder applies stacked T5 layers to produce contextualized hidden states, which are passed to the decoder. The decoder generates sequences conditioned on attention masks, producing multi-quantile forecasts. For extended horizons, decoding extrapolation is used. Finally, all predictions are rescaled using the stored normalization parameters.

### B.2 SPECIALIZED METHOD

**Timer-XL Liu et al. (2025a).** Timer-XL is a decoder-only Transformer designed for unified time series forecasting with long contexts. It generalizes next-token prediction from 1D to multivariate settings, enabling one model to handle univariate, multivariate, and covariate-informed tasks. The core innovation is TimeAttention, which disentangles fine-grained intra- and inter-series dependencies while preserving temporal causality and permutation-equivalence across variables. By enlarging the context to thousands of tokens, Timer-XL captures both local and global dynamics more effectively. Extensive experiments show state-of-the-art performance in supervised forecasting, covariate-informed prediction, and long-context benchmarks, while large-scale pre-training further yields strong zero-shot generalization, positioning Timer-XL as a versatile backbone for foundation models in time series.

**TimeXer Liu et al. (2024b).** TimeXer is a Transformer-based model that represents time series as sequences of patches. It employs a hierarchical design with patch embedding, temporal encoding,

and attention to capture both short- and long-term dependencies. To incorporate covariates, TimeXer introduces variate-level embedding, where external covariates are embedded and fused into the target series' patch representations. This integration enables the model to learn how external factors affect internal dynamics at the patch level, thereby improving predictions with exogenous information.

**iTransformer Liu et al. (2024a).**   iTransformer rethinks the role of Transformers in multivariate time series forecasting by inverting the architecture without altering native components. Instead of forming temporal tokens by merging variates at each timestamp, iTransformer treats each variate as a token. Self-attention is then applied to capture multivariate correlations, while feed-forward networks model series representations for each variate. This inversion mitigates issues of noisy attention maps and poor scalability in long lookback windows. Extensive experiments across diverse real-world datasets show that iTransformer achieves state-of-the-art performance, improves generalization to unseen variates, and scales effectively with longer histories, positioning it as a strong backbone for time series forecasting.

**PatchTST Nie et al. (2023).**   PatchTST converts time series into patches that serve as Transformer input tokens for forecasting. It relies on two key ideas: Patching, which segments series into subseries-level patches to capture local temporal patterns and reduce attention complexity for longer histories, and Channel Independence, which applies shared embeddings and Transformer weights to treat multivariate inputs as parallel sequences. Notably, PatchTST does not utilize covariate information in its predictions.

## C   HURST EXPONENT AND FRACTIONAL BROWNIAN MOTION

### C.1   HURST EXPONENT

The Hurst exponent Mandelbrot & Wallis (1969) is a widely used statistical metric for analyzing the long-term memory, self-similarity, and fractal characteristics of time series data. Originally introduced by H.E. Hurst in hydrology to study the Nile River's water levels, it has since been applied in diverse fields, including finance, geophysics, climate science, and signal processing.

The Hurst exponent quantifies the tendency of a time series to either persist in its trend, revert to the mean, or behave like a purely random process. Its values range from 0 to 1:

- $H = 0.5$ indicates a pure random walk, suggesting no long-term correlation between observations.
- $H > 0.5$ signifies persistence, meaning that increases (or decreases) are likely to be followed by further increases (or decreases), indicating long-range positive correlation.
- $H < 0.5$ implies anti-persistence, meaning that an increase is likely to be followed by a decrease and vice versa, indicating long-range negative correlation.

A common method for estimating $H$ is the rescaled range (R/S) analysis, which evaluates the variability of cumulative deviations from the mean relative to the standard deviation:

$$H = \frac{\log\left(\frac{R(n)}{S(n)}\right)}{\log(n)} \tag{9}$$

where $n$ is the window size, $R(n)$ is the range of cumulative deviations from the mean within the window, and $S(n)$ is the standard deviation of the data within the same window. By performing this calculation over multiple window sizes and applying regression on a log-log plot of $R(n)/S(n)$ versus $n$, the slope yields the Hurst exponent.

The Hurst exponent provides insight into the degree of long-range dependence in a time series, helping to distinguish between random, trending, and mean-reverting behaviors. This makes it a valuable tool for predicting future trends, assessing market volatility, modeling environmental processes, and analyzing any system exhibiting temporal correlations.

### C.2   FRACTIONAL BROWNIAN MOTION

Fractional Brownian Motion (FBM) Mandelbrot & van Ness (1968) is a continuous-time Gaussian process that generalizes standard Brownian motion by incorporating long-range dependence and

self-similarity. Introduced by Mandelbrot and Van Ness in 1968, FBM is widely used in fields such as finance, hydrology, geophysics, and signal processing to model time series exhibiting memory effects. An FBM process $B_H(t)$ is characterized by the Hurst exponent $H \in (0, 1)$, which determines the correlation structure of its increments. Its key properties include:

- **Self-similarity:** For any $a > 0$, $B_H(at) \overset{d}{=} a^H B_H(t)$, where $\overset{d}{=}$ denotes equality in distribution.

- **Stationary increments:** The increments $\Delta B_H(t) = B_H(t + \tau) - B_H(t)$ are stationary but generally correlated.

- **Covariance function:** The covariance between two points $t$ and $s$ is

$$\text{Cov}(B_H(t), B_H(s)) = \frac{1}{2}\Big(t^{2H} + s^{2H} - |t - s|^{2H}\Big). \tag{10}$$

The Hurst exponent $H$ determines the persistence of the process:

- $H = 0.5$ corresponds to standard Brownian motion with independent increments.

- $H > 0.5$ indicates persistence, where positive (or negative) increments are likely to be followed by increments of the same sign.

- $H < 0.5$ indicates anti-persistence, where positive increments are likely to be followed by negative increments, and vice versa.

FBM is particularly useful for modeling natural and financial processes with long-range correlations, allowing for the analysis and prediction of systems exhibiting temporal dependencies.

## D  EXPERIMENT DETAILS

### D.1  TRAIN-TEST SPLITTING

For all datasets, we reserve 10% of the total data as the test set to evaluate model performance. To generate multiple training and evaluation samples, we employ a sliding window approach, where a fixed-length window moves along the time series with a step size of 1. This allows the model to be tested on overlapping sequences and provides a more robust assessment of its forecasting ability. Additionally, within the training data, we further set aside a portion as a validation set to tune hyperparameters and prevent overfitting, ensuring that the model generalizes well to unseen data.

### D.2  EVALUATION METRICS

We evaluate the forecasting performance using four metrics: Mean Absolute Error (MAE) and Mean Squared Error (MSE) to assess point prediction accuracy.

**MAE**  MAE is a standard metric in time series forecasting that measures the average magnitude of errors between predicted values and actual observations, ignoring their direction. It provides a direct assessment of prediction accuracy. Formally, MAE is defined as:

$$\text{MAE} = \frac{1}{N}\sum_{i=1}^{N}|\hat{y}_i - y_i| \tag{11}$$

where $N$ denotes the total number of observations, $y_i$ represents the true value, and $\hat{y}_i$ is the predicted value at time step $i$. MAE computes the mean absolute deviation between predictions and ground truth, offering an interpretable metric that is less sensitive to large errors compared to squared-error-based measures.

**MSE**  MSE is a widely used metric in time series forecasting that quantifies the average of the squared differences between predicted values and actual observations. By squaring the errors, MSE

penalizes larger deviations more heavily, making it sensitive to outliers. Formally, MSE is defined as:

$$\text{MSE} = \frac{1}{N} \sum_{i=1}^{N} (\hat{y}_i - y_i)^2 \tag{12}$$

where $N$ denotes the total number of observations, $y_i$ represents the true value, and $\hat{y}_i$ is the predicted value at time step $i$. MSE provides a measure of overall prediction accuracy with an emphasis on larger errors.

