# OpenReview forum: "Filter before Plug: One-for-All Framework for Covariate-Aware Forecasting with Time Series Foundation Models"
_ICLR.cc/2026/Conference — ICLR 2026 Conference Withdrawn Submission_

### Official Review · Reviewer_SeXV · 2025-10-30

**Soundness:** 2
**Presentation:** 2
**Contribution:** 3
**Rating:** 4
**Confidence:** 4

**Summary:**

The paper proposes FLUG, a "One-for-All" framework designed to enhance Time Series Foundation Models (TSFMs) by incorporating exogenous covariate information in forecasting tasks. FLUG encompasses two independently trained modules: (1) an Endogenous Series Filter (EFit), guided by the Hurst exponent, that extracts the endogenous (self-dependent) components of a time series; and (2) a Covariate Plugin (CPin) using multi-scale patchification and causal-aware masking to capture exogenous influences from covariates. The paper demonstrates how these modules can be plugged into any TSFM without requiring end-to-end fine-tuning. Evaluation across standard and newly curated datasets suggests the modular approach consistently improves forecasting performance, especially in data-rich power generation environments.

**Strengths:**

1.	The independently trainable Filter and Plugin modules for TSFMs present a flexible approach to incorporating covariate information. This “One-for-All” design improves applicability over tightly coupled plugin approaches and end-to-end architectures.
2.	The CPin module includes a Gradient Reversal Layer for dynamic, data-adaptive causal masking, a nontrivial step beyond simple correlation-based selection. The effectiveness of this element is demonstrated empirically and visually.

**Weaknesses:**

1.	The paper repeatedly asserts that end-to-end integration of covariates undermines the intrinsic forecasting capability of pretrained TSFMs. However, no theoretical justification or controlled ablation supports this claim, leaving the argument largely anecdotal.
2.	The notion of “future-known” covariates is central to the proposed CPin module, yet its operational definition is unclear. For different datasets, the boundary between exogenous variables that are truly known ahead of time versus those only predictable via auxiliary models remains ambiguous.
3.	The experiments heavily rely on pretrained TSFMs such as TimesFM and Chronos, but the paper omits crucial details on how these models were obtained, what datasets they were trained on, and whether they were retrained or adapted. This lack of transparency hinders fair comparison and reproducibility.
4.	FLUG is only evaluated when coupled with pretrained TSFMs. It remains unclear whether the proposed “filter-before-plug” paradigm is compatible with or beneficial to fully end-to-end forecasting models such as Timer-XL, TimerXer, PatchTST, or iTransformer.
5.	The paper states that large-scale industrial datasets from Ningxia and Jilin are “processed and released,” but no download links, licenses, preprocessing pipelines, or privacy-handling protocols are provided. This lack of disclosure raises concerns about data accessibility and reproducibility.
6.	The additional EFit and CPin modules likely increase computational cost, yet the paper provides no quantitative analysis (e.g., FLOPs, inference latency, or parameter counts). It remains unclear whether the plugin overhead could outweigh performance gains, especially on smaller datasets or real-time applications.
7.	Although the paper claims that FLUG is a “One-for-All” paradigm, most datasets and case studies focus on energy forecasting. The absence of evidence in heterogeneous domains such as finance, healthcare, or commodity markets limits the generalizability of the claimed framework.

**Questions:**

See weaknesses

---

### Official Review · Reviewer_82Hr · 2025-10-30

**Soundness:** 2
**Presentation:** 1
**Contribution:** 2
**Rating:** 2
**Confidence:** 4

**Summary:**

This paper presents FLUG, a One-for-All framework for incorporating exogenous covariates into Time Series Foundation Models (TSFMs). It introduces two independently trained modules: EFit, which uses the Hurst Exponent to extract endogenous dependencies, and CPin, which applies multi-scale fusion and causal-aware masking to capture exogenous effects. By disentangling endogenous and exogenous components, FLUG can adapt to various TSFMs without fine-tuning. Experiments show that FLUG consistently improves forecasting performance, demonstrating its versatility and effectiveness.

**Strengths:**

1. The proposed One-for-All paradigm—where covariate and endogenous modules are trained independently of any foundation model—is conceptually elegant. It enables true plug-and-play adaptation across diverse TSFMs, significantly enhancing flexibility and reusability without the need for costly retraining or architectural coupling.

2. The release of large-scale photovoltaic and wind power datasets containing 1.3 billion samples with rich, well-documented covariate structures represents a valuable contribution to the time-series forecasting community. These datasets not only provide a realistic benchmark for covariate-aware forecasting but also offer an important resource for advancing research on scalable and generalizable TSFMs in real-world industrial settings.

**Weaknesses:**

1. The EFit and CPin modules are trained independently and later integrated into frozen TSFMs for inference. However, the paper does not clearly demonstrate how well these modules generalize when the TSFM representations differ from those seen during their own training. In practice, the assumption of full independence between the modules and the foundation model may not hold, potentially limiting robustness and adaptability across architectures.

2. Although ChronosX is briefly discussed, the paper lacks direct quantitative or ablation comparisons with other plugin-based adaptation methods under equivalent training budgets or data settings. Such comparisons would be essential to fairly evaluate the claimed “One-for-All” advantage of FLUG over existing plugin paradigms.

3. The paper provides no clear analysis of the computational or latency overhead introduced by integrating both EFit and CPin into existing TSFM pipelines. A discussion of inference speed, additional memory usage would help assess the framework’s practicality in real-world forecasting systems.

4. The writing quality, particularly in the methodology section, is notably poor. The descriptions often omit key details such as module input–output specifications, mathematical formulations, tensor dimensions, and intermediate representations. As a result, the method is unnecessarily difficult to follow, making it challenging for readers to fully reproduce or verify the proposed approach.

**Questions:**

1. Covariates from other modalities, such as images or text, can also have a significant impact on time series prediction. Can the method proposed in this paper generalize to these types of data?

---

### Official Review · Reviewer_qMhU · 2025-10-31

**Soundness:** 3
**Presentation:** 3
**Contribution:** 3
**Rating:** 4
**Confidence:** 3

**Summary:**

This paper introduces FLUG, a One-for-All framework designed to integrate exogenous covariates into Time Series Foundation Models (TSFMs) without retraining or fine-tuning foundation models. It separates time series into endogenous and exogenous components via two independently trained modules, EFit for long-term dependency extraction, and CPin for modeling causal, multi-scale covariate effects

**Strengths:**

- **Innovative modular design**. FLUG introduces independently trainable filter and plugin modules that can be seamlessly attached to various pretrained TSFMs, establishing a general “one-for-all” adaptation paradigm.
- **Theoretically grounded endogenous filtering.** The use of the Hurst Exponent and Fractional Brownian Motion-based loss provides a statistically meaningful way to isolate temporally dependent (endogenous) components from noisy or covariate-influenced signals.
- **Extensive and rigorous evaluation.** Experiments cover both public benchmarks (e.g., GFC14, PDB) and newly released industrial datasets, with comprehensive ablations confirming the contributions of each module and loss component.

**Weaknesses:**

1. **Questionable motivation of the Hurst exponent.** In line 213, the paper assumes that a Hurst exponent greater than 0.5 implies strong trends and periodicity. However, the interpretation of the Hurst exponent is conceptually inaccurate. As acknowledged by the authors in Appendix C.1, a Hurst value greater than 0.5 indicates increases are likely to be followed by further increases. This property is not related to periodicity at all. Therefore, using the Hurst exponent as an indicator to extract periodic components in the endogenous series lacks theoretical rigor, and given the importance of periodicity in time series, I have doubts about the representative learning ability of this method.

2. **Lack of discussion on zero-shot forecasting capability.** One of the central motivations for TSFMs is their strong zero-shot generalization, yet the paper provides no analysis of whether FLUG preserves or diminishes this property. Since both the EFit and CPin modules introduce additional learned parameters trained on downstream datasets, a dedicated study or ablation on zero-shot transferability is crucial, especially when the paper mainly focuses on ''one for all''.

3. **FBM-based generator may be overly idealized.** FBM assumes Gaussianity and stationary increments. Many real-world series violate these assumptions. The paper should justify why FBM is a valid anchor for endogenous structure and does not harm forecasting ability.

4. **Inconsistent trend in multi-scale patchify analysis.** In Figure 4, model performance exhibits a clear trend of deterioration when extending the scales from the second scale ([16, 32]) to the fourth scale ([4, 8, 16, 32]). The paper does not discuss this anomaly. A more thorough explanation would strengthen the empirical analysis.

5. **Unclear implementation details of mask computation.**
The description of the causal-aware masking mechanism lacks clarity, particularly regarding whether Pearson and Spearman coefficients are computed across the entire time series or within local temporal windows. Without a rigorous experimental protocol, it is difficult to assess whether the reported correlations truly reflect dynamic causal relationships between covariates and targets.

**Questions:**

See weaknesses

---

### Official Review · Reviewer_gwAV · 2025-11-01

**Soundness:** 2
**Presentation:** 2
**Contribution:** 2
**Rating:** 4
**Confidence:** 3

**Summary:**

Authors propose a framework to enable covariate-aware forecasting using TSFMs.

This adaptation is achieved through two modules

1. Endogenous Series Filter:- this module captures stable patterns and dependencies from the history. This module is a TS encoder which takes as input the original series and constructs a new series which is further filtered using a discriminator to only retain the endogenous information
2. Covariate Plugin:- this module divides covariates into patches at different scales and retains the covariates with the largest influence on the target time series.

Output from both modules are added together to obtain the final output. These modules are trained using a combination of loss functions.

Most TSFMs are pretrained to handle univariate forecasting. The main contribution of the work is the framework to perform covariate-aware forecasting with TSFMs.

**Strengths:**

1. Authors have performed a detailed ablation studies.
2. The proposed framework is novel and ablation studies clearly show the significance of each component.

**Weaknesses:**

1. Evaluation seems incomplete. EPF datasets contain 5 time series (EPF-PJM, EPF-BE, EPF-DE, EPF-FR, EPF-NP). Not sure why evaluation is restricted to only one time series (PJM)
2. Proposed framework is not tested for long-term forecasting.
3. Computational overhead of the framework is not discussed.
4. Training details are not included. Particularly the size of the encoders which can introduce the largest computation overhead.
5. The solution is great, but it takes away the ability of foundation models to forecast zero-shot, as the framework requires training for each dataset.
6. Evaluation with ChronosX (Arango et al.) is missing.

**Questions:**

1. What is the size and the architecture of the encoders used in EFit and CPin?
2. What is the training and computation overhead of this framework?
3. Why are other EPF datasets excluded?

---

### Note · Authors · 2025-11-28

I have read and agree with the venue's withdrawal policy on behalf of myself and my co-authors.